# Targeting pancreatic cancer with combined inhibition of EGFR and RAF

Jakob Brandstetter[1], Lea Goldstein[1], Tim Schreiber[1], Rupert Palme [2], Tobias Lindner [3], Markus Joksch[3], Bernd Krause[4], Brigitte Vollmar[1], Simone Kumstel [1*]

**1** Rudolf-Zenker-Institute of Experimental Surgery, University Medical Center Rostock, Rostock, Germany, **2** Department of Biological Sciences and Pathobiology, Unit of Experimental Endocrinology, University of Veterinary Medicine, Vienna, Austria, **3** Core Facility Multimodal Small Imaging, University Medical Center Rostock, Rostock, Germany, **4** Department of Nuclear Medicine, University Medical Center Rostock, Rostock, Germany

* simone.kumstel@uni-rostock.de

## Abstract

Pancreatic cancer is the third leading cause of cancer-related death, with a 5-year survival rate of only 10%. Preclinical studies remain essential for identifying novel therapeutic strategies, discovering biomarkers, and deepening the understanding of disease biology. The most frequent driver mutation in pancreatic cancer is the G12D mutation in the KRAS gene, present in approximately 90% of the tumors. A recent study demonstrated complete regression of KRAS-driven pancreatic cancer upon systemic ablation up- and downstream signaling proteins EGFR and C-RAF. Building on these findings, we investigated the therapeutic benefit of combining the EGFR inhibitor erlotinib with the novel pan-RAF inhibitor LXH-254. The anticancer effects of this combination were assessed *in vitro* in murine and human pancreatic cancer cell lines by evaluating cell proliferation, cell death and phosphorylation of key signaling proteins. Subsequent *in vivo* studies were performed in an orthotopic murine pancreatic cancer model and in genetically engineered KPC mice, using daily oral administration of LXH-254 (35 mg/kg) and erlotinib (75 mg/kg). While the treatment robustly inhibited MAPK signaling and caused significant anti-proliferative effects *in vitro,* it did not improve survival or reduce tumor burden in either *in vivo* model. hese results contrast with previous reports of efficacy from monotherapies in xenograft models, highlighting the limitations of current preclinical approaches. Our findings underscore the need to develop more effective pathway-targeted inhibitors, and preclinical models that predict clinical outcomes more accurately.

## 1. Introduction

Pancreatic ductal adenocarcinoma (PDAC) is characterized by a 5-year survival rate of just 10% and currently ranks as the third deadliest cancer, projected to become second by 2030 [1–4]. Resection of the tumor remains the only curative therapy, yet

**Data availability statement:** All relevant data are within the maniscript and its Supporting Information files.

**Funding:** This study was funded by the German Research Foundation (KU36301-1). The funder had no role in study design, data collection and analysis, decision to publish, or preparation of the manuscript.

**Competing interests:** The authors have declared that no competing interests exist.

many patients are diagnosed at a stage when the tumor has either already locally advanced or metastasized to distant sites [5]. Thus, early tumor detection through sensitive screening systems is desirable but the discovery of effective therapeutic approaches is also urgently needed.

Even for patients eligible for surgery, additional chemotherapy, administered either before (neoadjuvant) or after (adjuvant) surgery, is commonly used to reduce the risk of disease recurrence. Treatment with FOLFIRINOX or gemcitabine/nab-paclitaxel are established chemotherapeutic-regimens [6,7] to prolong the survival of patients but severely limited by resistance mechanisms and adverse side effects [8,9].

Genomic analyses revealed that most PDAC patients share a phenotype of common driver mutations [10] and has enabled subtyping of patients according to molecular profiles of the tumor mass and microenvironment, which are associated with differences in histopathology and ultimately therapeutic responses, and clinical outcome between each other [11–13].

Among these driver alterations, mutations in *KRAS*, most prominently $KRAS^{G12D}$, occur in about 90% of PDAC cases and drive constitutive KRAS activation, promoting tumor cell proliferation [14]. Inhibition of downstream pathways such as MAPK/ERK or PI3K/AKT/mTOR has been explored in PDAC and other solid tumors, including colorectal carcinoma (CRC) and non-small cell lung cancer (NSCLC), leading to implementation of several targeted agents into therapeutic regimens in the clinic [15–18]. This targeted approach can achieve effective anti-cancer effects with fewer side effects, as supported by a recent study demonstrating combining chemotherapy with targeted agents provided superior quality of life in CRC patients versus chemotherapy alone [19].

As previously shown in a preclinical study the systemic ablation of both EGF receptor (EGFR) and C-RAF led to a complete regression of KRAS positive PDAC in a murine model [20]. The therapeutic inhibition of EGFR in combination with the ablation C-RAF led also to a strong regression of PDAC [20] indicating that these proteins might be essential for the development of PDAC. A targeted inhibition of both targets might therefore represent a promising approach for therapeutic treatment in PDAC in the future, though effective inhibition of C-RAF remains challenging and the development of reliable C-RAF inhibitors is considered crucial for the PDAC therapy.

Recently LXH-254 (Naporafenib), a novel type II small-molecule kinase inhibitor of RAF, has been developed and shown to exert anticancer activity by inducing the enzymatically inactive DFG-out conformation, with high preference for the kinases C-RAF and B-RAF [21–24]. As the biocompatibility of LXH-254 was demonstrated in a clinical trial [25,26] utilizing this specific RAF-inhibitor appeared promising as new potent pan-RAF inhibitor.

In this preclinical study we investigated the effect of a new combination therapy comprising LXH-254 together with EGFR inhibitor erlotinib, a small molecule tyrosine-kinase inhibitor that competitively targets the intracellular domain of EGFR and which is FDA-approved for treatment of patients with advanced pancreatic cancer in combination with gemcitabine [16,27].

We first tested this new combination therapy in an *in vitro* setting on different pancreatic cancer cell lines and assessed its effect on cell proliferation, viability and the phosphorylation status of EGFR and downstream MAP/ERK signalling proteins, followed by further investigation of this dual targeting in a syngeneic, murine orthotopic PDAC model and in genetically engineered KPC mice.

## 2. Materials and methods

### 2.1. Cell culture and *in vitro* assays

The murine Panc02 cells were provided by the national cancer institute (Frederick, Maryland, USA) and cultured in RPMI 1640 medium (PAN-Biotech, Aidenbach, Germany) with 10% fetal calf serum and 1% penicillin/ streptomycin. The murine 6606PDA were originally provided by Prof. Tuveson (University of Cambridge, UK) and the human cell line MiaPaca2 was purchased from ATCC (Manassas, VA, USA), both cell lines were cultured in DMEM medium plus 10% fetal calf serum and 1% penicillin/ streptomycin.

The therapeutics LXH-254 and erlotinib (MedChem Express, Monmouth Junction, NJ, USA) were diluted in 100% DMSO to a 10 mM stock solution for the *in vitro* assays. To quantify the tumor cell proliferation 1–2 x $10^3$ cells were plated out in 96-well plates and treated after 24 h with distinct concentrations of LXH-254 plus 10 or 15 µM erlotinib for 72 h. Exemplary pictures after 72h of treatment were taken with a microscope (Leica DMI 4000B, Wetzlar Germany). The proliferation was quantified by performing the BrdU assay according to manufacturer's instructions (11647229001, Roche Diagnostics, Rotkreuz, Switzerland). Cell death was quantified by the assessment of trypan blue positive cells. 1 x $10^4$ cells were plated out on a 24-well plate, incubated at 37 °C for 24 h and treated with 4 µM LXH-254 and 10 µM erlotinib for 72 h. The cells were treated with trypsin, blinded and trypan blue was mixed with a ratio of 1:1. The percentage of necrotic cells was quantified under the microscope.

To quantify the inhibition of the EGFR and the MAPK/ERK pathway, the phosphorylation of the proteins was quantified via SDS page and western blot. 2 x $10^5$ Panc02 cells were cultured for 24 h in a 6-well plate. Afterwards the cells were treated with the distinct concentrations of LXH-254, erlotinib or DMSO for 24h or the indicated time periods in the figures (0.5 h, 1 h, 6 h and 24 h). The proteins (30 µg for EGFR/pEGFR and ERK/pERK, and 20 µg for MEK/pMEK) of the cell lysates were further separated in SDS polyacryl gels and transferred to a polyvinyl difluoride membrane (Immobilon-P; Millipore, Eschborn, Germany), as previously described [28]. To quantify the signaling proteins the membranes were blocked with 2.5% BSA and incubated overnight at 4 °C with primary antibodies at a dilution of 1:1000 for EGFR (D38B1, Cell Signaling Technology, Danvers, Massachusetts, USA), MEK (47E6, Cell Signaling Technology) and ERK (9102, anti-p44/42, Cell Signaling Technology). On the next day the blots were incubated for 1 h with the secondary antibody (7076, anti-mouse-IgG HRP-linked, Cell Signaling Technology, 1:10.000). Protein expression was visualized by luminol-enhanced chemiluminescence (ECL plus; GE Healthcare, Munich, Germany) and digitalized with Chemi-Doc XRS System (Bio-Rad Laboratories, Munich, Germany). All blots were stripped and incubated over night at a dilution of 1:1000 of the antibodies detecting the phosphorylated form of the signaling proteins pEGFR (2234, Tyr1086, pEGFR, Cell signaling Technology), pMEK (9121, Ser217/221, pMEK, Cell signaling Technology), pERK (9102, p44/42, Erk1/2, Cell Signaling Technology) and protein detection was conducted as described as above. The membranes were stripped again afterwards and treated with an antibody against β-actin (Sigma Aldrich). The quantitative analysis of the protein bands was further performed with the program ImageLab (Bio-Rad, Hercules, California; USA).

### 2.2. Animals

Breeding pairs of C57BL/6J were originally purchased from Charles River Laboratories (Wilmington, Massachusetts, USA). Breeding pairs of KPC mice (Kras^tm4Tyj^Trp53^tm1Brn^Tg(Pdx1-cre/Esr1*)#Dam/J), were purchased from Jackson Laboratories (Bar Harbor, Maine, USA). The animals were bred under specific pathogen-free conditions in the main animal facility of the University Medical Center Rostock. Health monitoring of the mice was routinely performed according to

FELASA guidelines. In the last two years, the following pathogens were detected in the facility *Helicobacter sp.*, *Rodentibacter pneumotropicus* and *Murine noro virus* in some mice, these mice were not used for further experiments. All animal experiments were approved from the local authority (Landesamt für Landwirtschaft, Lebensmittelsicherheit und Fischerei Mecklenburg-Vorpommern; 7221.3-1-010/21). During the experiment, the mice were single housed in type III cages (Zoonlab GmbH, Castrop-Rauxel, Germany), in a 12-h dark-light cycle, at 21±2 °C and a relative humidity of 60±20%. The mice had free access to pellets (10 mm, ssniff-Speziadiäten GmbH, Soest, Germany) and tap water. Enrichment was provided with nesting material (shredded tissue paper, Verbandmittel GmbH, Frankenberg Germany), paper role (75×38 mm, ssniff-Spezialdiäten) and wooden stick (40×16×10 mm, Abedd, Vienna, Austria).

### 2.3. Cell-derived pancreatic cancer model

Metamizole (3 mg/ml) was applied as continues analgesia daily in the drinking water up to 10 days before tumor cell injection until euthanasia of the mice. C57BL6/J mice (12 male and 13 female) between 12–26 weeks old were anesthetized with 3–5 vol.% isoflurane in a chamber. The mice were placed on a heating plate at 37 °C and anesthesia was continued at 1–2.5 vol.% isoflurane over a mask. The eyes were kept wet by eye ointment. Carprofen (5 mg/kg, Rimadyl®, Pfizer GmbH, Berlin; Germany) was injected subcutaneously as additional pain medication. The abdomen of the mice was shaved, disinfected, and opened via laparotomy. Panc02 cells (5 µl, $1x10^4$ cells 1:1 matrigel and PBS) were injected into the pancreas. The peritoneum was closed with a continuous suture (vicryl 5–0, Johnson & Johnson Medical GmbH, New Brunswick, NJ, USA), afterwards the epidermis was closed by single knots (prolene 5–0, Johnson & Johnson Medical GmbH). One week after tumor cell injection, the mice were randomized into the different treatment groups. The mice were treated either with LXH-254 (35 mg/kg, MedChemExpress, Monmouth Junction, NJ, USA), erlotinib (75 mg/kg, MedChemExpress, Monmouth Junction, NJ, USA), the combination of both therapeutics, or the vehicle solution (90% PEG400 (Merck KGaA, Darmstadt, Germany) and 10% Tween80 (Sigma-Aldrich, St. Louis, USA)) by oral gavage once a day, five times a week with a maximum of 30 doses. The dosing and the treatment duration was chosen in accordance with previous studies indicating a moderate inhibition of tumor progression and no toxicity in different rodent cancer models [29,21,30]. After randomization of the animals, 3 male and 3 female mice were treated with the vehicle and 3 male and 3 female mice received LXH-254. Erlotinib was administered to 3 male and 4 female mice and the combinatorial treatment was conducted in 3 male and 3 female animals. According to the previously published score sheet [31], the mice were euthanized as soon as relevant health related criteria were noticed with body weight decrease (10%) and abnormal body posture being the most frequent observations. The first mouse had to be euthanized 20 days after tumor cell injection and the last mouse at day 65.

### 2.4. Genetic pancreatic cancer model

KPC dams received oral tamoxifen (6 mg; Sigma-Aldrich, dissolved in corn oil) on the day of birth and on days 1, 2, and 4 postpartum, resulting in pup exposure to tamoxifen via maternal milk. After 4–5 weeks the pups were separated from the mother. *In vivo* imaging via MRT started at the age of 75 days every two weeks. As soon as pancreatic tumors were detected by MRI, the therapeutic intervention was started, as described above.

### 2.5. Assessment of welfare parameters

During the experiment, the body weight was assessed daily and the mice were checked for the occurrence of health-related criteria according to our previously published score sheet [31,32]. The staff was well-trained for assessing the score sheet criteria and health related symptoms. The other distress parameters were monitored on distinct time points during the experiment: before and directly after tumor cell injection, at the recovery days 1–3, as well as during the early (day 8), middle (day 22) and late (day 38) treatment period. Perianal temperature was measured three times in the perianal region of the mice with a contactless infrared thermometer (WEPA Apothekenbedarf GmbH & Co. KG, Hilscheid,

 

Germany) and the mean value was calculated. Burrowing behavior was quantified by placing a burrowing tube (6.5 x 6.5 x 15 cm), filled with 200 g pellets, around 3 pm (± 0.5 h) into the home cage of each mouse. 8 animals which burrowed less than 100 g during 2 h and 3 animals which burrowed less than 150 g overnight before tumor cell injection were excluded from the burrowing analysis. Two hours later at 5 pm the burrowed amount of pellets was calculated, as well as on the next morning between 8–10 am [33,34]. Nesting behavior was assessed by placing a cotton nestlet (5 cm square, ZOON-LAB GmbH, Castrop-Rauxel, Germany) between 5 and 6 pm into the home cage. The nest was scored on the next morning via a 1–5 point score from Deacon et al. [35] with an additional score of 6 that was applied for a perfect nest, if 90% of the circumference of the nest wall was higher than the body height of the mouse. 9 out of 25 mice that created a nest scored less than a score of 4 before tumor cell injection were excluded from the nesting assessment. The mouse grimace scale (MGS) was monitored by placing each mouse into a polycarbonate box (9 x 5 x 5 cm). The box was placed into an illuminated tent and was additionally enlightened from the front. After 5 minutes of acclimatization, the mouse was filmed for 5 minutes with a digital single-lens reflex camera (Canon EOS 70D, Tokyo, Japan). Three pictures were captured from each mouse from the video at each time point, randomized, blinded and scored by three researchers according to Langford et al. [36]. The bodyweight and the distress parameters were assessed daily to ensure continuous health monitoring of the mice. As soon as health related criteria such as ruffled fur, 10% body weight loss or hunched posture were scored according to the score sheet [31,32], the welfare parameters MGS, perianal temperature, burrowing and nesting were assessed daily as well, to establish early humane endpoints for the murine models. The monitoring of the mouse grimace scale during this time was restricted to a short observation in the home cage by three different researchers. All mice were euthanized, by cervical dislocation under isoflurane anesthesia (4 vol.%), when one of the humane endpoint criteria such as 15% body weight loss, short spasms, temporary paralysis symptoms, abnormal respiratory sound or breathing, body condition score of 2, pronounced apathy, squeaking due to pain, self-mutilation, ascites or bent posture were recognized. Bent posture or 15% bodyweight loss was scored most frequently as humane endpoint criteria. From all mice, used in this study (25 C57BL/6J and 6 KPC mice), no mouse died on its own; all animals were euthanized painlessly before they suffered significantly. 20 mg/ml BrdU was injected intraperitoneal 2 hours before euthanasia. The data collected during the humane endpoint phase has already been published to address the scientific question of early humane endpoint determination, with the goal of sustainably reducing the suffering of mice in future studies [32].

For the quantification of fecal corticosterone metabolites (FCMs), the cage was replaced 24 h before feces collection. The feces were dried at 65 °C for 4 h and stored at −20 °C. 50 mg of dry feces were extracted with 1 ml 80% methanol. Afterwards a 5α-pregnane-3β,11β,21-triol-20-one enzyme immunoassay was performed as described before [37]. Blood samples were taken by retro-orbital puncture during a short isoflurane anesthesia (4 vol.%). Blood count was quantified by VETSCAN® HM5 (Abaxis, Inc., Union City, California, USA). The plasma was separated after centrifugation (1200 × g for 10 min at 6 °C) and stored at −80 °C. Plasma corticosterone concentrations were measured using an ELISA Kit (DEV9922, Demeditec Diagnostics GmbH, Erfurt, Germany) according to manufacturer's manual. For histological analysis tumors were fixed with 4% paraformaldehyde, sliced in 4 µm sections, and dried at 65 °C for 2h. Afterwards the slides were stained with hematoxylin and eosin (H/E) to quantify necrotic areas in the tissue. BrdU positive cells were detected with anti-BrdU (1:50; MA5–12502 Thermofischer, Waltham, Massachusetts, USA) with secondary detection by HRP-conjugated (polyclonal goat anti-mouse, 1:100 Dako, Germany Hamburg) or anti-phospho Stat3 (Tyr705) (#9145 Cell Signaling Technology, Danvers; Massachusetts, USA) followed by secondary antibody (goat anti rabbit IgG 1:200, 97048 abcam, Cambridge, United kingdom). Cell death was quantified by Apop-Tag® Plus Peroxidase *In Situ* Apoptosis detection Kit (S7101 Millipore, Billerica, Massachusetts, USA). For immunostainings up to six images per tumor were taken at 200x magnification using a microscope (Axioscop2 7, Zeiss Microscopy, Oberkochen, Germany). Positive cell staining or stained area was quantified by Qupath v.6.0 [PMID: 29203879]. For H&E staining, the entire tumor sections were scanned at 5x magnification and percentage of necrotic area was quantified using Qupath v.6.0.

 

### 2.6. *In vivo* Imaging

PET-CT imaging was performed with the radiotracer [$^{18}$F]FDG to quantify the glucose metabolism of the tumors at two different time points during therapeutic treatment as previously described [31]. The mice were anesthetized with 1.0–2.5 vol.% isoflurane, and eye ointment was applied. Mice were intravenously injected with ~ 15 MBq of [$^{18}$F]FDG into the tail vein. 1 h after injection, static PET scans in head-prone position were measured for 15 min (Inveon PET-CT Siemens, Knoxville, TN, USA). During the procedure, the body temperature of the mice were kept stable via heating pad. The PET-image reconstruction method comprised a two-dimensional ordered subset expectation maximization (2D-OSEM) algorithm with four iterations and six subsets. Attenuation correction was achieved with whole body CT scan and a decay correction. PET images were corrected for random coincidences, dead time, and scatter. The uptake of $^{18}$F-FDG (%injected dose per g body weight) was quantified in the tumor as metabolic volume (30% of the hottest voxel), by using the software Inveon Research Workplace (Siemens Healthcare AG, Zurich, Switzerland). The MRI imaging was performed as described previously [38]. The KPC mice were scanned with 7T MRI (magnetic resonance imaging, BioSpec 70/30, 7.0 Tesla, gradient insert: BGA-12S HP Bruker BioSpin GmbH, Ettlingen, Germany), combined with a volume resonator (86 mm inner diameter) and a 2 x 2 receive surface coil. Mice were anesthetized with 1.0–2.5 vol.% isoflurane and scanned with morphological T2 weighted TurboRARE (Rapid Acquisition with Relaxation Enhancement) sequences with following parameters: TE/TR 25/1880 ms; FoV:approx. 40 x 28 mm, matrix: 200 x 200; voxel size: 0.2 x 0.14 mm, slice thickness 1 mm, 25 slices. Therapeutic interventions were initiated once a tumor was detected by MRI.

### 2.7. Statistics and manuscript preparation

Data was graphed and analyzed with the program GraphPad Prism 8.0 (GraphPad Software, San Diego, USA). Dose response curves of the distinct cell lines indicate mean ± SEM. The *in vitro* comparison between all the treatment groups, as well as the *in vivo* data were depicted as box plots, indicating median, 25 and 75 percentiles and 95% confidence interval as whiskers. The protein expression was graphed as bar chart indicating mean ± 95% confidence interval. Survival curves were depicted as Kaplan-Meier curves and statistically evaluated using log-rank test. Data normality was assessed using the Shapiro-Wilk test ($\alpha = 0.05$). Normally distributed data were analyzed with one-way ANOVA, two-way ANOVA, or a mixed-effects model followed by Tukey's or Dunnett's post hoc test, while non-normal data were processed with the Kruskal-Wallis test and Dunn's multiple comparisons.

Some parts of this manuscript were revised and optimized for clarity and conciseness with the assistance of OpenAI's ChatGPT (GPT-5, August 2025 Version). The authors reviewed and verified all generated content for accuracy.

## 3. Results

### 3.1. Erlotinib combined with LXH-254 inhibits tumor cell proliferation and leads to cell death *in vitro*

First, we investigated the effect of the reagents on cell proliferation by BrdU assay in an *in vitro* setting utilizing the murine pancreatic cancer cell lines Panc02. Across different concentrations from 0.5 µM to 32 µM, both LXH-254 monotherapy and the combination with 10 µM erlotinib led to a dose-dependent inhibition of cell proliferation in Panc02 cells. For 72 hours of treatment, an IC$_{50}$ value of 5.95 µM was calculated for LXH-254 monotherapy and 2.25 µM in combination erlotinib (Fig 1A). The comparison of the combination therapy to single-agent and vehicle treatment indicated a significantly stronger inhibition of proliferation for the combination therapy (Fig 1B). Trypan blue staining revealed significantly higher rates of cell death following the dual therapeutic approach compared to either monotherapy or vehicle control (Fig 1C). Exemplary images of the Panc02 cells after 72h treatment indicating the strong anti-proliferating effect of LXH-254 and erlotinib (Fig 1 D).

Similar results were obtained with the 6606PDA cell line. Dose–response curves demonstrated an IC$_{50}$ of 4.53 µM for LXH-254 monotherapy and 1.18 µM for the combination therapy (S1A Fig). Dual inhibition also resulted in significantly reduced cell proliferation compared to monotherapy and vehicle (S1 A-C Fig).

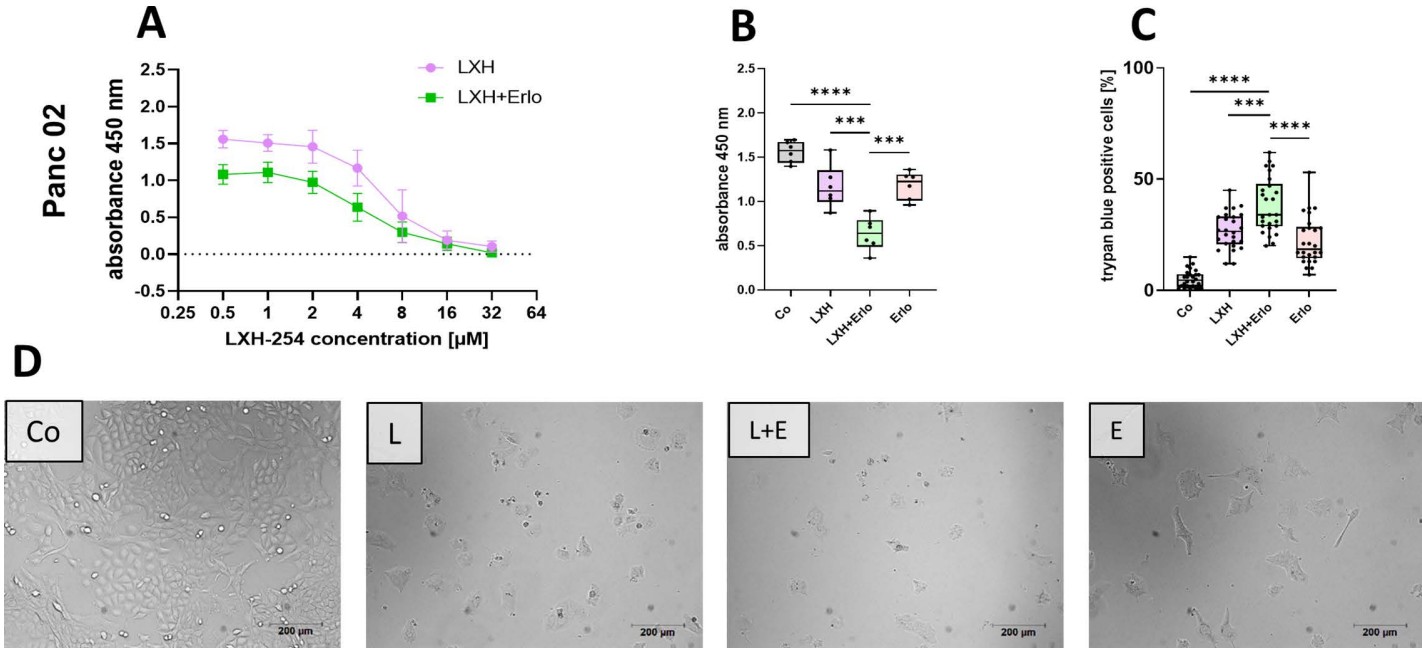

**Fig 1. Anti-cancer effects of the combination therapy on the murine pancreatic tumor cell line Panc02 by quantifying cell proliferation and cell death after 72 h.** Dose-dependent inhibition of murine Panc02 cell proliferation was quantified either after monotherapy with LXH-254 or after combination therapy with 10 µM erlotinib **(A)**. Differences in tumor cell proliferation **(B)** and cell death rates **(C)** were investigated in Panc02 cells treated with vehicle (Co), 4 µM LXH-254 (LXH), 10 µM erlotinib (Erlo), or combination therapy (LXH+Erlo; **B**). Exemplary images of Panc02 cells treated for 72 h with either vehicle (Co), 4 µM LXH **(L)**, in combination with 10 µM erlotinib (E + L) or monotherapy of 10 µM erotinib (E; **D**). Statistics were carried out using one-way ANOVA with Tukey-test for multiple comparisons. *** p ≤ 0.001, **** p ≤ 0.0001; A, B: n = 6; C: n = 26.

Comparable effects were observed in the human MIA PaCa-2 cells, with $IC_{50}$ values of 3.78 µM for LXH-254 monotherapy and 1.20 µM for the combination therapy (S1D Fig). Moreover, the combination treatment significantly inhibited cell proliferation compared to vehicle control and both monotherapies (S1 E-F Fig).

### 3.2. Treatment with mono- and combination therapy leads to inhibition of the phosphorylation cascade in the signalling pathway

To evaluate the influence on intracellular signalling pathways in Panc02 cells, we quantified the ratios of phosphorylated to non-phosphorylated forms of EGFR, MEK, and ERK proteins via western blot analysis. For this purpose, time-dependent incubations with 1 µM LXH-254 were performed, ranging from 0.5 to 24 hours. A significant decrease in the pEGFR/EGFR ratio was only detected after 24 hours of incubation, compared to the control treated cells (Fig 2A). The pMEK/MEK ratio was already significantly reduced from 1–6 hours (Fig 2B). A significant decline in the pERK/ERK ratio was observed after 1 and 3 hours of treatment (Fig 2C).

Equivalent time-course experiments were conducted using 1 µM erlotinib. A significant inhibition of the activated EGFR protein, compared to control-treated cells was detected via significant reduced pEGFR/EGFR ratio between 0.5 and 6 hours of treatment (Fig 2D). A significant inhibition of MEK phosphorylation was noted after 3 hours and again after 24 hours (Fig 2E). The pERK/ERK ratio was significantly reduced after 1 hour and continued to decrease with longer incubation times (Fig 2F).

We next evaluated concentration-dependent effects of both erlotinib and LXH-254 monotherapies (0.1 µM, 1 µM, 10 µM) during 24 h treatment in comparison to the control. A significant decrease in the pEGFR/EGFR ratio was observed

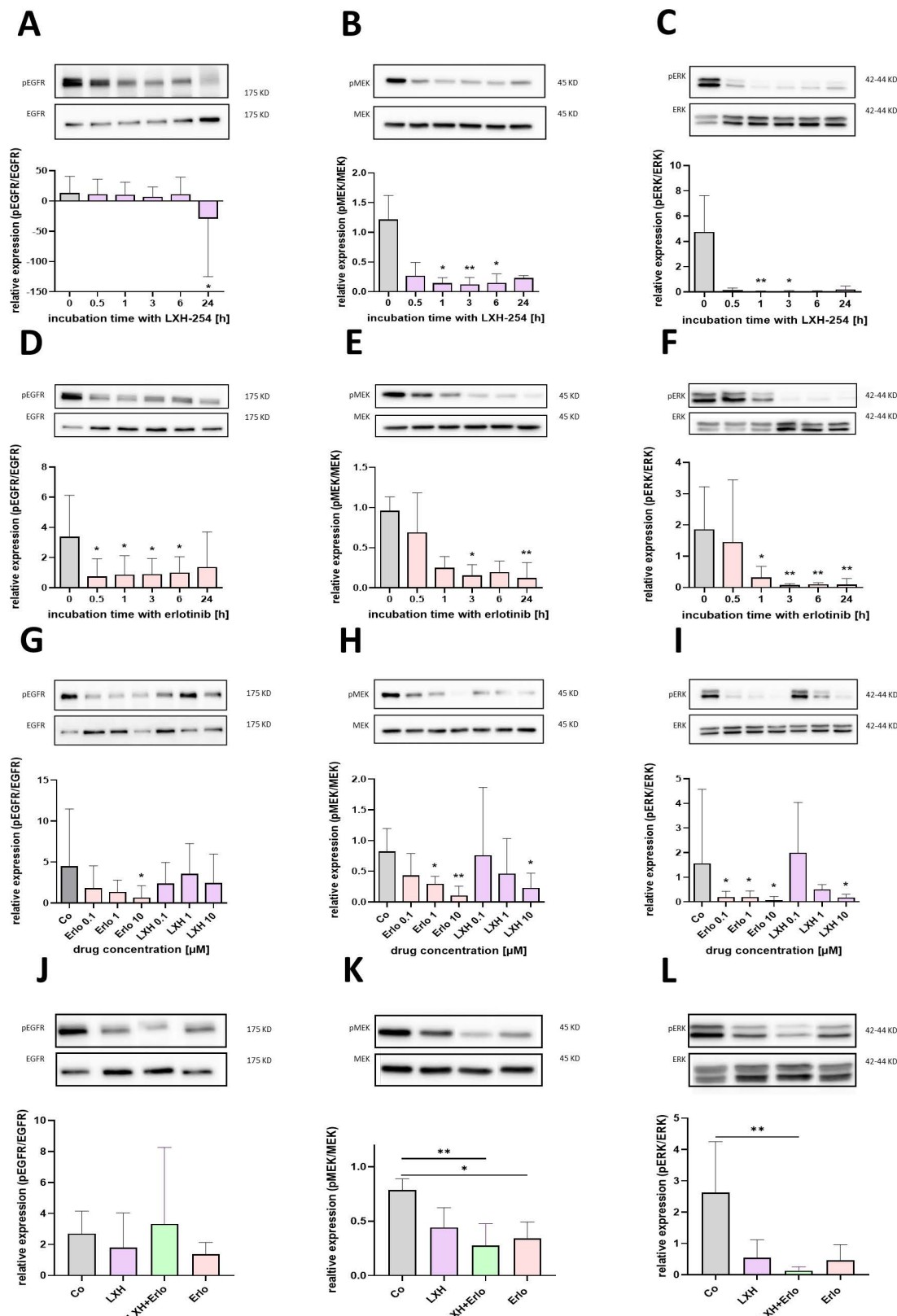

**Fig 2. Analysis of phosphorylation of EGF-receptor and MAPK-signaling pathway after therapy of Panc02 cells with LXH-254 and erlotinib.**
Ratio between phosphorylated and non-phosphorylated protein of epidermal growth factor receptor (EGFR) **(A)**, MEK **(B)**, and ERK **(C)** after incubation

with 1 µM LXH-254 for 0.5, 1, 3, 6 and 24 hours. Quantification of phosphorylated protein expression of EGFR **(D)**, MEK **(E)** and ERK **(F)**, after treatment with 1 µM erlotinib for the indicated time points. Dose dependent inhibition of signal protein phosphorylation were quantified for varying concentrations of both agents and a treatment duration of 24 hours **(G-I)**. The inhibition of protein phosphorylation of EGFR **(J)**, MEK **(K)** and ERK **(L)** was quantified after treating the cells with vehicle (Co), 4 µM LXH-254 (LXH), 10 µM erlotinib (Erlo) or the combinatorial treatment for 24 hours. Statistics were carried out using one-way-ANOVA or mixed-model. * $p \leq 0.05$, ** $p \leq 0.01$; A-C: n = 4; D-F: n = 4; G-I: n = 3; J-L: n = 5-6.

following treatment with 10 µM erlotinib, but not for the different LXH-254 concentrations ( Fig 2G). The phosphorylation of MEK was significantly reduced when treated with 1 µM and 10 µM erlotinib, as well as at 10 µM LXH-254 (Fig 2H). Significant attenuation of ERK activation was observed at all three erlotinib concentrations, and at the highest concentration of LXH-254 (Fig 2I).

Finally, we compared the phosphorylation status across the mono treatments and the combination therapy for EGFR, MEK, and ERK after 24 h for 4 µM LXH-254, 10 µM erlotinib, and the combination of both. No significant differences were observed for the pEGFR/EGFR ratio (Fig 2J). However, the pMEK/MEK ratio was significantly reduced for the erlotinib monotherapy and combinatorial treatment compared to the control (Fig 2K). Notably, the pERK/ERK ratio was significantly decreased in combination therapy-treated cells compared to the control (Fig 2L).

### 3.3. Erlotinib combined with LXH-254 fails to produce an anticancer effect in an orthotopic murine model

Based on the promising *in vitro* results of the combinatorial therapy, its potential anticancer effects were further evaluated *in vivo*. A short survival time was observed in the combination therapy group (median survival of 36.5 days). In contrast, mice in the vehicle group survived significantly longer (median of 45.5 days; Fig 3A-D). Additionally, no significant reduction in tumor mass was detected in the dual therapy group compared to the other groups, neither at the individual humane endpoint of each mouse (Fig 3E-F), nor at the specific time points during the experiment as indicated by PET-CT imaging and quantification of the metabolic tumor volume (30% of the hottest voxel of injected dose per g body weight; Fig 3G).

### 3.4. Multiparametric distress evaluation suggest that animals adapt to therapeutic regime, with signs of distress during early and late phases of tumor progression

Longitudinal assessment of welfare parameters revealed a slight initial increase in distress (S2 Fig). Distress then subsided during the middle phase of tumor progression but rose again in the late phase, correlating with higher tumor burden. Similar patterns were observed in behavioural measures such as burrowing and nesting activity (S3 Fig). Haematological analysis during treatment revealed a downregulation of lymphocytes with a significant reduction for control-treated mice. Neutrophil counts were upregulated in the early treatment phase in LXH-254 receiving animals. No significant differences between the treatment groups were quantified for the other haematological parameters (S4 Fig).

### 3.5. Combination therapy with erlotinib and LXH-254 leads to no significant therapeutic effect in KPC mice

In addition to the investigations in the cell-based orthotopic mouse model, the comparison of the combination therapy and vehicle treatment was evaluated in KPC mice. No significant differences of the overall survival were observed between the two treatment groups ($median_{vehicle}$: 84 days, $median_{LXH+Erlo}$: 94 days; S5A-B Fig). Interestingly, mice treated with the combination therapy exhibited significantly larger tumors compared to the vehicle treated mice at the individual endpoints (S5C Fig).

The tumors from all treatment groups in the C57BL/6J mice were further processed for immunohistological analysis. The percentages of necrotic area, proliferating cells, apoptotic cells and pSTAT3-positive area were quantified (Fig 4A–D). No significant differences were observed among the treatment groups for any of these parameters, including necrotic area, cell proliferation, apoptosis, or pSTAT3 positivity. The same immunohistological analysis were performed on

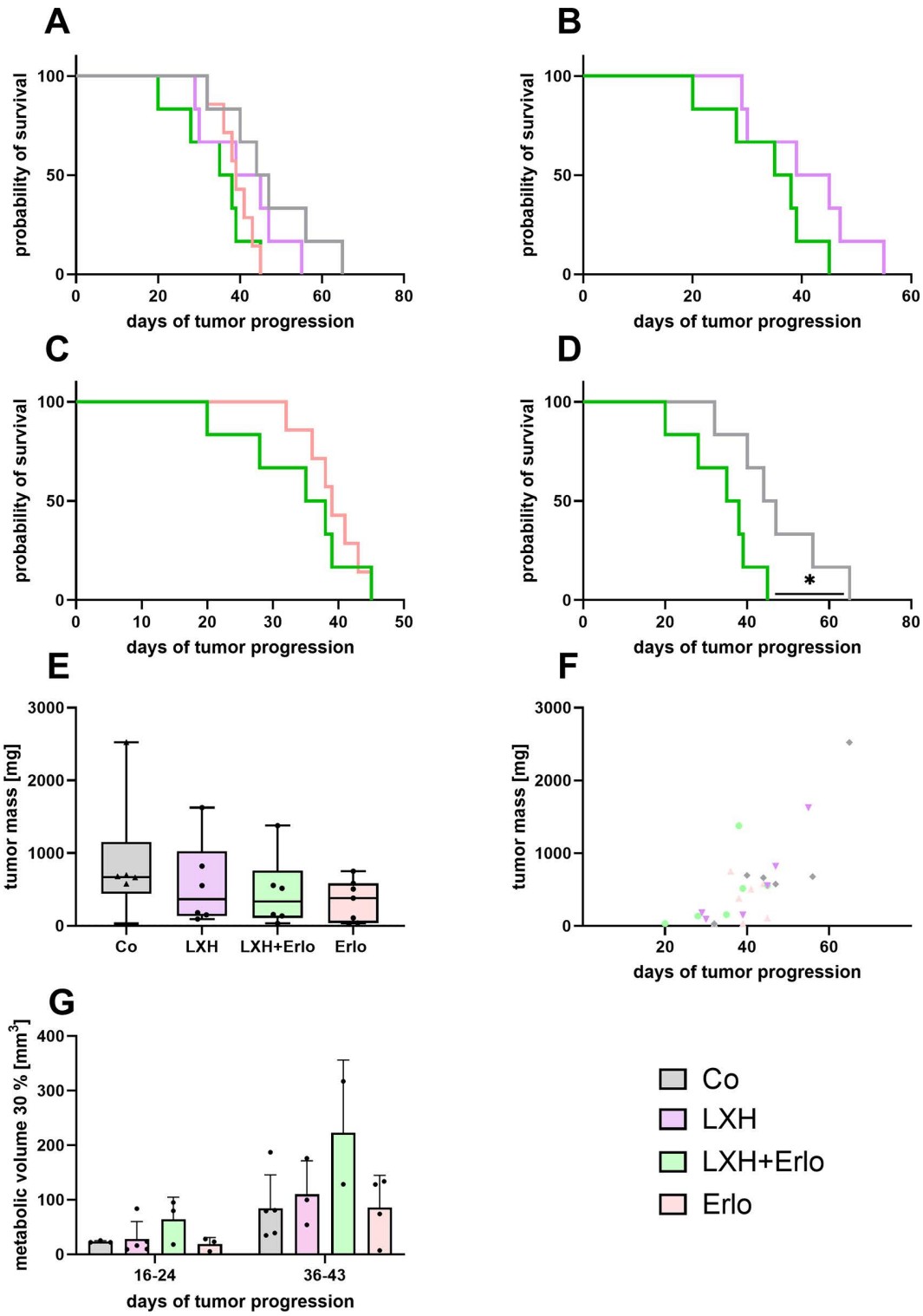

**Fig 3. Analysis of survival and tumor progression in an orthotopic murine pancreatic model following combinatorial treatment with LXH-254 and erlotinib.** Probability of survival was plotted for each treatment group after injection of tumor cells and oral therapy **(A-D)**. Averaged tumor weights are displayed for each treatment group **(E)** and temporal representation according to the time point of euthanasia **(F)**. Metabolic tumor volume (30% of the hottest voxel of injected dose per g body weight) was determined by [$^{18}$F]FDG uptake at two time points during the treatment period **(G)**. Statistics were carried out using one-way ANOVA or Kruskal-Wallis test. * $p \leq 0.05$; A-F: LXH; LXH+Erlo; Co n=6; Erlo n=7; G: Erlo, LXH+Erlo n=6; LXH, Co n=; LXH+Erlo; Co n=5.

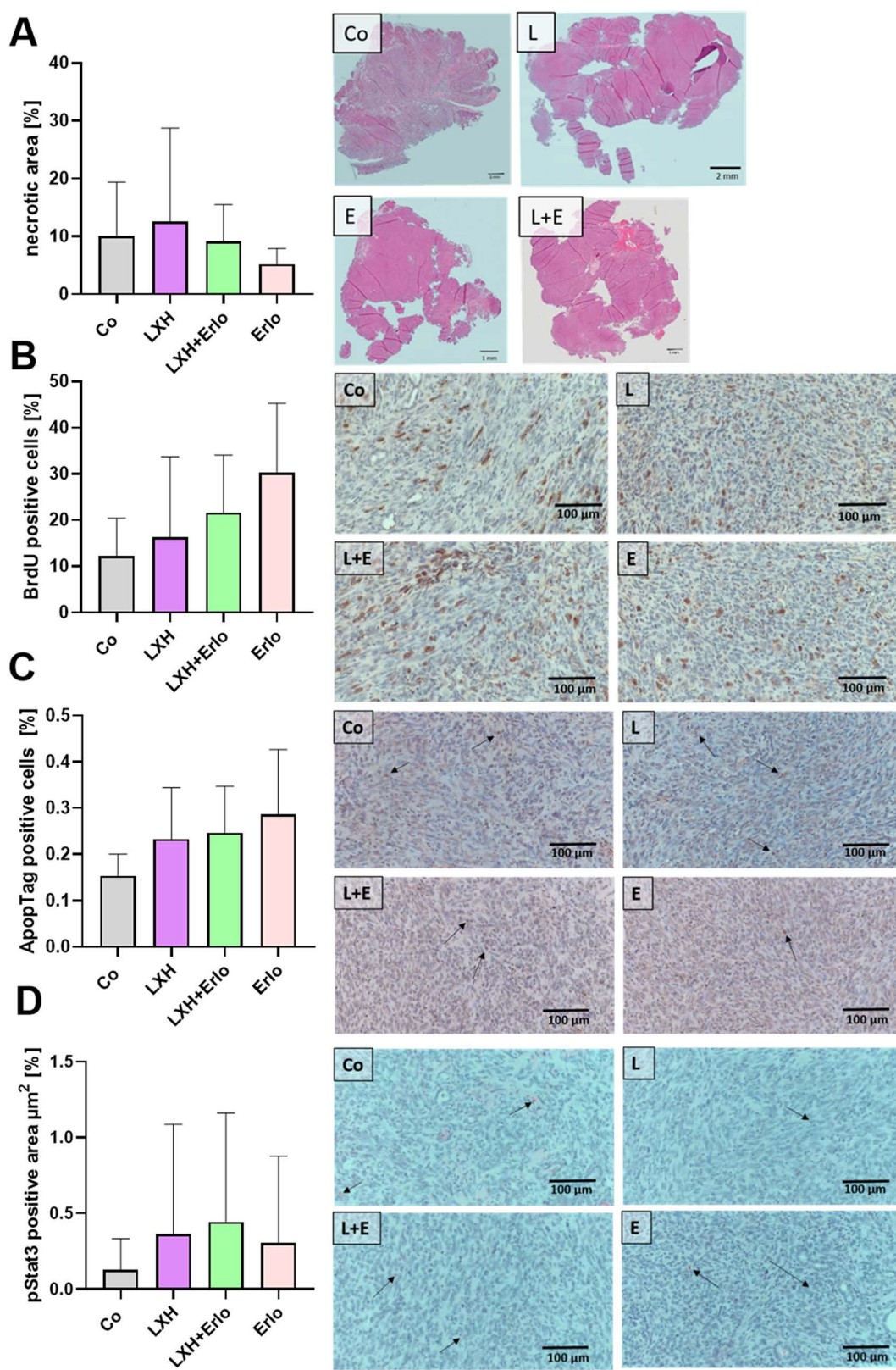

**Fig 4. Immunohistological analysis of tumors from treated C57BL/6J mice.** Quantification of percentages of necrotic area via H/E staining **(A)**, proliferating cells by BrdU incorporation **(B)**, apoptotic cells via ApopTag® Peroxidase **(C)** and pSTAT3 positive area **(D)**, respectively for control, LXH-254,

LXH-254 plus erlotinib, or erlotinib treated mice. All staining's are illustrated in one exemplary picture for each treatment group. Arrows marking the specific stained cells, or area. Statistics were carried out using one-way ANOVA with Tukey-test for multiple comparisons. Co: n = 5-6, LXH: n = 3-6, LXH+Erlo n = 3-4, Erlo n = 4-7.

the tumors from the KPC mice and the two different treatments. No significant different were observed either in necrotic area, proliferating or ApopTag positive cells, nor in pSTAT3 positive area (S6 Fig).

## 4. Discussion

In their preclinical study, Blasco et al. demonstrated that a combined ablation of EGFR and RAF led to a complete regression of pancreatic tumors in mice [20]. These findings suggested that a dual administration of FDA-approved EGFR-inhibitor erlotinib and the new RAF-inhibitor LXH-254 might harbour the potential for an effective treatment of PDAC. Upon incubation with LXH-254 monotherapy for 72 h we calculated $IC_{50}$ of 5.95 µM and 4.26 µM for the murine cell lines Panc02 and 6606PDA. An $IC_{50}$ of 3.78 µM for the human cell line MIA-Paca2 corresponded to the calculations of Monaco et al. in their studies, where they determined an $IC_{50}$ value of 2.16 µM [26]. Erlotinib monotherapy led to a significant inhibition of cell proliferation at concentrations of 10 µM and 15 µM on all three pancreatic cancer cell lines. Dose–response curves and $IC_{50}$ calculations were not performed, as erlotinib concentration was selected based on its effect in combination with varying LXH-254 concentrations. Upon comparison of the monotherapies to combined treatment with erlotinib and LXH-254 our data revealed a significantly stronger effect on proliferation and cell death upon dual treatment (Fig 1, S1 Fig). When evaluating the impact of the drugs on the intracellular signalling cascades a significant reduction of the activated signalling proteins was observed for certain incubation times and concentrations of LXH-254, erlotinib or the combination therapy (Fig 2). Lange et al. also detected an inhibitory effect on ERK phosphorylation when investigating the effect of erlotinib on the two pancreatic cancer cell lines BxPC-3 and Capan-1 [39]. This inhibitory effect on the activated pERK protein on BxPC-3 as well as on Panc02 was also demonstrated in another study after exposing EGF-activated cells to increasing concentrations (5, 10 and 20 µM) of erlotinib [40]. A clear reduction of the pMEK/MEK and pERK/ERK ratios was also noted when murine PDAC cells K375 and K399 were treated with concentrations of 1 and 10 µM erlotinib [41]. These findings are consistent with our results, which demonstrated that erlotinib inhibits the MEK/ERK signalling pathway in pancreatic cancer cells and confirm its concentration-dependent inhibitory effect on distinct PDAC cell lines. For RAF-inhibitor LXH-254 Monaco et al. demonstrated an inhibitory effect on pMEK and pERK at 1 µmol/L LXH254 in human MIA-Paca2 cells after one hour [26]. This is also in line with our findings where we could demonstrate a significant decline in both pMEK and pERK in murine Panc02 cells after one-hour exposure of LXH-254 as well as a distinct inhibitory effect on proliferation at a concentration of 1 µM LXH-254.

Based on the promising *in vitro* results, the combination treatment was subsequently evaluated in a syngenic orthotopic pancreatic cancer model in mice. However, the promising therapeutic effect of LXH-254 in combination with erlotinib observed *in vitro* did not translate *in vivo* as neither prolonged overall survival nor significant inhibition of tumor progression was achieved in two murine models, even when compared to control (Fig 3 and S5 Fig). It should be noted that the number of mice used in the KPC model was very low restricting the validity and statistic power of the data. However, due to the results further increasing of the sample size for the sake of data would be unethical. In addition to tumor weight and survival, immunohistological analysis of the tumors, including assessment of necrotic area as well as proliferating and apoptotic cells also failed to reveal a detectable therapeutic effect of the treatment.

Therefore, we did not quantify anti-cancer effects of the combinatorial treatment. In contrast, mice treated with LXH-254 and erlotinib exhibited significantly reduced survival compared with vehicle-treated controls (Fig 3D). Which might indicate a possible toxic effect of the combinatorial treatment. We can not exclude subtle or organ-specific toxic effects, as we did not perform extensive clinical chemistry or histopathological toxicology analyses. However, when assessing animal

distress in a multiparametric fashion, no severe toxic side effects were quantified for the combination treatment (S2–S3 Figs). At the beginning of the therapy mice exhibited a slight increase in distress as evidenced by a decrease in body weight, elevated MGS scores and the increased plasma corticosterone and FCM concentrations. We interpreted these primarily as general adaption responses to oral gavage as previously described alongside mild side effects of the drugs [42–44]. During the mid-phase of therapy, only minor signs of distress were observed as indicated by slight increases in MGS scores and blood corticosterone concentrations. These findings suggest a generally acceptable tolerability for this combination therapy targeting EGFR and RAF and confirm the low side effects profile observed previously upon systemic ablation of EGFR and RAF, in which the initial weight loss reversed after a few weeks [20]. In the late phase of therapy, an increased impairment of wellbeing was observed for all treatment groups, reflected in elevated distress scores and higher FCM concentrations, probably due to an increased tumor burden. Behavioural assays, including overnight burrowing and nesting activity, revealed similar trends with impairments mainly observed during the early and late phases of therapy. These findings are consistent with our previous findings (S3 Fig) [42]. In addition to the low side effect profile quantified by the welfare parameters, we did not quantify severe changes of the blood counts except for downregulation of lymphocytes during the therapeutic intervention, especially in control-treated mice (S4 Fig). However, lymphocytopenia is frequently observed during the progression of solid tumors and represents an adverse prognostic factor [45,46]. In addition, the neutrophils were significantly upregulated in LXH-254-treated animals (S3 Fig). Importantly, no mouse required euthanasia due to suspected treatment related toxicity. The endpoint criteria were body weight loss (>15%) and hunched body posture which occurred as consequence of tumor progression since most animals (17 out of 25; 68%) exhibited invasive tumor growth into the intestine. Based on these observations, we refrain from classifying the combinatorial treatment as directly toxic. However, the treatment was associated with accelerated tumor progression and increased intestinal invasion and should therefore be considered harmful in this model and under the used dosing conditions.

In the orthotopic *in vivo* study, erlotinib monotherapy (75 mg/kg, 5 days/week for max. 30 days) resulted in a reduction of the tumor mass by 66% compared to vehicle-treated mice but did not translate into prolonged survival. Erlotinib is FDA-approved for the treatment of non-small cell lung cancer and advanced pancreatic cancer, in the latter case in combination with gemcitabine [16]. Previous studies have reported pronounced antitumor effects in several xenograft models, where erlotinib monotherapy induced distinct tumor shrinkage compared to control groups [47–50]. Notably, Yamamura et al. demonstrated that a 10-day regimen of daily i.p. injections of 10 mg/kg body weight significantly reduced tumor volume and extended survival in subcutaneous BxPC-3-tumor-bearing mice [50]. However, all these experiments were conducted in severely immunocompromised mice, which are generally more sensitive to therapeutic intervention. As a result, such xenograft models often display stronger responses to treatment than immunocompetent mouse models, limiting their predictive value for clinical outcomes [50].

For LXH-254 monotherapy (35 mg/kg, 5 days/week for max. 30 days) no therapeutic effect could be observed *in vivo* in the present study. Contrary to our results, Ramurthy et al. have demonstrated the efficacy of LXH-254 using the same concentration of 35 mg/kg in a Calu-6 xenograft nude rat model upon oral administration with dose-dependent anti-tumor activity and acceptable tolerability as indicated by only moderate body weight loss and no signs of toxicity or increased mortality [21]. Tumorstatic effects upon applying up to 100 mg/kg LXH-254 on a daily basis were observed in another preclinical study utilizing several xenograft models, with MAPK driven tumor cells for melanoma, ovarian carcinoma, colon cancer and pancreatic cancer [26]. As described above the absence of any therapeutic effect in our *in vivo* model might derive from principal differences of model organisms where particularly the presence of an intact immune system significantly alters a host's response towards therapeutic intervention [51]. Given the presence of an intact immune system, the syngeneic and the genetically engineered murine cancer models may offer a greater translational relevance compared to the xenograft models [52]. This statement is supported by results from a clinical trial in which patients with advanced solid tumors harboring MAPK-signalling pathway alterations (KRAS-positive non-small cell lung cancer and NRAS-positive melanoma) were treated with LXH-254, demonstrating an acceptable biosafety profile but only limited antitumor activity [25].

 

Our results, together with the above-mentioned clinical trial, suggest that RAF inhibition by LXH-254 may not be sufficiently potent to achieve effective anticancer activity *in vivo*. Therapeutic targeting either of pan-RAF or selective isoforms (ARAF, BRAF or CRAF) have been under thorough investigation in diverse cancer entities, but RAF-inhibitors that efficiently translate into the clinic remain lacking [53]. Computational models by Sevrin et al. suggested that combining type I½ RAF-inhibitors (e.g., encorafenib) with type II RAF-inhibitors (e.g., TAK-632) could synergistically inhibit ERK signalling, a prediction later validated experimentally in different cell lines harboring distinct KRAS-mutations [54]. These insights underscore the need for ongoing research and further *in vivo* testing of novel RAF inhibitors to achieve a durable inhibition of RAF *in vivo*.

Aside from the limited inhibitory effect of LXH-254, the underlying reasons for the significantly reduced survival observed in the cell-derived murine model–and an increased tumor weight in the genetically engineered KPC mice following combinatorial treatment–remain unclear. While our western blot analyses revealed that essential members of MAPK-signalling cascade were successfully inhibited after incubation with therapeutics up to 24 h, a compensatory effect could still be assumed after longer time in an *in vivo* setting. One important resistance mechanism after combined RAF and EGFR ablation was introduced from Blasco et al. by an increased activation of JAK-STAT signalling in resistant tumors [20]. We did not quantify a significant increase of pSTAT3-positive area after combinatorial treatment of LXH-254 and erlotinib in both pancreatic cancer models (S5D and S6DFig.). Therefore, we cannot conclude that increased activation of JAK-Stat signaling pathway accounts for the observed therapeutic resistance. Another notable resistance mechanism after treatment with early-generation EGFR tyrosine kinase inhibitors such as erlotinib, gefitinib or afatinib is the development of T790M mutation. This mutation was observed in 60% of patients with acquired EGFR inhibitor resistance leading to alterations in drug binding and enzymatic activity of the mutated EGFR receptor, ultimately preventing the competitive binding of the inhibitors [55]. LXH-254 is reported to inhibit both BRAF and CRAF, which theoretically should prevent the paradoxical MAPK activation by RAF-dimerization [56–58]. However, as noted before, we assume that the *in vivo* potency of LXH-254 as inhibitor of BRAF and CRAF is likely very limited and its inhibitory activity against ARAF has been described as markedly lower *in vitro* [26,58]. In addition, its efficacy would be likely reduced in tumours harbouring novel oncogenic RAF-mutations (e.g., L485F, L505H) that stabilize the kinase's regulatory spine element, a change correlated with increased resistance towards both first- and second-generation RAF inhibitors [59]. Recently Sanclemente et al. used mouse strains that expressed knock-in alleles of two kinases-dead RAF1 variants (RAF1D468A, RAF1K375M) to demonstrate that the therapeutic benefit of RAF-1 ablation in *KRAS/TRP53* mutant lung tumors is not mediated by inhibition of its kinase catalytic activity, but rather by the loss of its anti-apoptotic functions [60]. RAF-1 has been shown to interact with pro-apoptotic the kinases ASK1 and MST2 [61,62], thereby suppressing apoptotic signalling. Consequently, instead of directly targeting of RAF1 kinase activity, future therapeutic strategies should focus on disrupting its anti-apoptotic, either by blocking its interaction with ASK1 and MST2 or by suppressing RAF-1 expression using selective RAF-1 degraders [60].

Combined inhibition of EGFR and RAF can lead to selective pressure within the tumor and force the emerging resistant sub clones such as KRAS-mutant cells which might bypass upstream inhibition [63]. For example, oncogenic KRAS can bypass the MAPK-axis and drive tumor cell proliferation via phosphoinositide 3-kinase (PI3K) pathway [64]. In its GTP-bound form, RAS can directly bind the p110α catalytic subunit, activating AKT and mTORC1, which suppress pro-apoptotic factors and promote and drive cell cycle progression, compensating for the loss of ERK signalling [65]. Targeting a different member of MAPK-pathway, Mirzoeva et al. showed that small-molecule inhibition of MEK could also activate PI3K-pathway in breast cancer cells, a feedback mechanism that is disrupted by additional PI3K targeting [66]. Increased PI3K activity and hypermethylation of several tumor suppressors were also described as mechanisms in CDK4/RAF1 resistant lung adenocarcinomas [67]. Moreover, BRAF inhibition in CRC is linked to Wnt/β-catenin upregulation through focal adhesion kinase (FAK) activation proposing another bypass route to boost tumor cell proliferation [68]. Hence, these alternative pathways can hereby compensate for

therapeutic targeting of MAPK-signalling. Beyond that, combined EGFR and BRAF inhibition might also reinforce the pro-cancer effects of the tumor microenvironment in PDAC by reprogramming stromal cells, as observed in melanoma cells, where paradoxical activation of fibroblasts can trigger integrin β and FAK-mediated signalling, promoting ERK activation and tumor cell survival [69].

Ongoing studies target diverse signalling pathway axes to achieve stronger and longer-lasting antitumor effects. A phase I trial is currently evaluating the ERK inhibitor ulixertinib in combination with the cyclin-dependent inhibitor palbobiblib, aiming to counteract upstream MAPK pathway rewiring suppress cell proliferation during the G1- to S-phase transition (clinical trials.gov NCT03454035). The recent FDA approval of the dual RAF/MEK inhibitor avutometinib with the FAK inhibitor defactinib for KRAS-mutated recurrent low-grade serous ovarian cancer underlines a first success for this multi-pathway targeting approach [70]. This specific combination is now being tested alongside gemcitabine and nab-paclitaxel in previously untreated PDAC patients (clinical trials.gov NCT05669482).

Ultimately, testing this new combination therapy in the present study highlighted the general challenge of translating *in vitro* effects into *in vivo* efficacy. Our study illustrates that a marked reduction in tumor cell proliferation and viability under controlled cell culture conditions does not necessarily reflect anticancer activity within a living organism. Taken together, our study demonstrated a strong anti-cancer potential *in vitro*. However, these effects were not translatable into therapeutic efficacy in the cell-induced and genetic murine PDAC models used in this study. This underscores the need to employ multiple animal models including at least one with an intact immune system to generate results that can be translated into clinical application.

## Supporting information

**S1 Fig. Anti-cancer effects of the combination therapy on the murine and human pancreatic tumor cell lines by quantifying cell proliferation and cell death after 72 h.**
(PDF)

**S2 Fig. Multi-parametric evaluation of distress in an orthotopic murine pancreatic model during combination therapy.**
(PDF)

**S3 Fig. Analysis of behavioral parameters for welfare assessment in an orthotopic murine pancreatic model during combination therapy.**
(PDF)

**S4 Fig. Blood count before and during the therapy of all treatment groups.**
(PDF)

**S5 Fig. Quantification of therapeutic efficacy of combinatorial treatment with LXH-254 and erlotinib in a genetic pancreatic cancer model (KPC).**
(PDF)

**S6 Fig. Immunohistological analysis of tumors from treated KPC mice.**
(PDF)

**S7 Fig. Raw images.**
(PDF)

**S1 File. Raw data file.**
(XLSX)

## Acknowledgments

The authors are grateful for the perfect technical assistance from Anne Rupp, Eva Lorbeer, Christin Schlie, Johanna Förster, Berit Blendow, Dorothea Frenz, Karin Gerber, Illona Klamfuß.

## Author contributions

**Conceptualization:** Simone Kumstel.

**Formal analysis:** Jakob Brandstetter, Simone Kumstel.

**Funding acquisition:** Simone Kumstel.

**Investigation:** Jakob Brandstetter, Lea Goldstein, Tim Schreiber, Rupert Palme, Tobias Lindner, Markus Joksch, Simone Kumstel.

**Methodology:** Jakob Brandstetter, Rupert Palme, Simone Kumstel.

**Project administration:** Bernd Krause, Brigitte Vollmar, Simone Kumstel.

**Supervision:** Bernd Krause, Brigitte Vollmar, Simone Kumstel.

**Visualization:** Jakob Brandstetter, Simone Kumstel.

**Writing – original draft:** Jakob Brandstetter, Simone Kumstel.

**Writing – review & editing:** Jakob Brandstetter, Simone Kumstel.

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
