## [Decision Letter · Decision Letter 0]

28 Oct 2025

Dear Dr. Kumstel,

Thank you for submitting your manuscript to PLOS ONE. After careful consideration, we feel that it has merit but does not fully meet PLOS ONE’s publication criteria as it currently stands. Therefore, we invite you to submit a revised version of the manuscript that addresses the points raised during the review process.

We look forward to receiving your revised manuscript.

Kind regards,

Yoshihiro Sowa

Academic Editor

PLOS ONE

Journal Requirements:

2. To comply with PLOS One submissions requirements, in your Methods section, please provide additional information regarding the experiments involving animals and ensure you have included details on (1) methods of sacrifice, (2) methods of anesthesia and/or analgesia, and (3) efforts to alleviate suffering.

“This study was funded by the German Research Foundation (KU36301-1)”

4. Please note that funding information should not appear in the Acknowledgments section or other areas of your manuscript. We will only publish funding information present in the Funding Statement section of the online submission form. Please remove any funding-related text from the manuscript.

Reviewers' comments:

Reviewer's Responses to Questions

**Comments to the Author**

1. Is the manuscript technically sound, and do the data support the conclusions?

Reviewer #1: Yes

Reviewer #2: Yes

2. Has the statistical analysis been performed appropriately and rigorously?

Reviewer #1: Yes

Reviewer #2: Yes

3. Have the authors made all data underlying the findings in their manuscript fully available?

Reviewer #1: Yes

Reviewer #2: Yes

4. Is the manuscript presented in an intelligible fashion and written in standard English?

Reviewer #1: Yes

Reviewer #2: Yes

Reviewer #1: The paper of Brandstetter et al. mainly shows negative in vivo therapeutic results, validating the previous data of Blasco et al., that obtained therapeutic effect by genetic but not by pharmacologic strategies against EGFR and Raf1. The work merits publication if they need to include the following information:

• The fact that in vivo survival is lower under the treatment regimen (erlotinib and LXH-254) suggest that the in vitro results were not therapeutic but toxic. This information should be included in the discussion.

• Studies in Barbacid´s lab using different RAF kinase inhibitors (panRAF inhibitors MLN2480 and LSN3074753, a RAF1 selective inhibitors GW5074, and the paradox breaker PLX8394), have failed to show significant therapeutic activity on Kras/Trp53 mutant lung and PDAC tumors. The reason for this lack of therapeutic activity has been unveiled by the studies that have demonstrated that expression of two independent kinase dead isoforms of RAF1 (RAF1K375M and RAF1D468A) failed to show any relevant anti-tumor activity, thus indicating that the therapeutic activity of RAF1 observed in GEM tumor models of lung and pancreatic cancer is not mediated by its kinase activity, a truly unexpected result (Esteban-Burgos L, et al. Proc Natl Acad Sci U S A. 2020;117(39):24415-26). Instead, the results suggest that the anti-tumor activity observed upon RAF1 ablation is due to the elimination of its anti-apoptotic activity (Sanclemente M, et al. Cancer Cell. 2021;39(3):294-6.). This information should be included in the discussion or introduction.

• One of the resistant mechanisms described in Blasco et al. and recently in https://doi.org/10.1101/2025.08.04.668325 is the activation of STAT3. The authors should compare the levels of pSTAT3 in the tumors treated in vivo with vehicle or with erlotinib and LXH-254.

Reviewer #2: Authors Brandsetter et al. have presented findings that demonstrate a lack of therapeutic benefit from the combined inhibition of EGFR and RAF in two robust in vivo models, despite strong in vitro efficacy. In this reviewer's opinion, this "negative" finding makes an important contribution to the field of pancreatic cancer research. The reviewer enthusiastically agrees with the authors' conclusions, which advocate for the development of more effective pathway-targeted inhibitors and more accurate preclinical models. This is a direct, evidence-based recommendation for the research community.

The authors have appropriately selected strong KPC and orthotopic models that are universally accepted for their preclinical study aimed at dual inhibition, which is based on a sound scientific rationale. However, despite the clear evidence of in vitro work and laboratory experiments, there are several aspects that need clarification and further elaboration:

1. The in vivo work presented lacks any morphological assessment, which is not reported in the methodology or results. The reviewer would like to see changes in H&E morphology, as well as in situ molecular analyses, such as immunohistochemistry (IHC) or RNA in situ hybridization. Specifically, this reviewer is interested in IHC work that could demonstrate reduced apoptosis, increased senescence, alterations in key proliferation markers, p-ERK levels, or the presence of immune cells. Such evidence would provide solid, in situ support for the authors' conclusion regarding a lack of tumor regression. This omission is critical, as documenting morphological and IHC evidence is standard practice in the preclinical field. Representative images showcasing vehicle treatment, monotherapies, and combined therapy, alongside relevant protein expression IHC work, would be beneficial.

2. The Methods section should include more detail on how the drug doses were selected (e.g., based on maximum tolerated dose, previous literature, or pharmacokinetics/pharmacodynamics studies). This information is essential for reproducibility.

3. The Discussion should be expanded to hypothesize about the exact mechanisms underlying the in vivo failure. Possible factors to consider include whether the in vivo environment (e.g., stroma, immune cells, drug metabolism/pharmacokinetics) interfered with drug efficacy, or if the inhibition was simply insufficient given the complexities of the tumor microenvironment. Again, a morphological assessment conducted by an experienced pathologist, along with IHC evaluations, would provide valuable insight. I strongly recommend that the editorial board of PLOS One consider adding a co-author who is a pathologist to lend further credibility to the negative findings- if the authors request a change in authorship during revision phase.

4. While I commend the authors for conducting a scientifically rigorous study, I recommend that they complete the work, as it currently feels unfinished. I suggest enhancing the figure panels by including gross tumor images that illustrate the sizes of tumors across the various treatment groups in comparison to the vehicle group, as well as in vitro images, rather than relying solely on bar plots and Kaplan-Meier plots. This would greatly improve the overall appeal and meaning of the study.

I am happy to re-assess this manuscript and look forward to see the next draft of authors.

.

Reviewer #1: No

Reviewer #2: No

---

## [Author Response · Author response to Decision Letter 1]

22 Jan 2026

We answered the comments of the reviewers in deatail in the attached PDF document Response to Reviewer.pdf.

---

## [Decision Letter · Decision Letter 1]

17 Feb 2026

Dear Dr. Kumstel,

Thank you for submitting your manuscript to PLOS ONE. After careful consideration, we feel that it has merit but does not fully meet PLOS ONE’s publication criteria as it currently stands. Therefore, we invite you to submit a revised version of the manuscript that addresses the points raised during the review process.

We look forward to receiving your revised manuscript.

Kind regards,

Yoshihiro Sowa

Academic Editor

PLOS One

Journal Requirements:

Reviewers' comments:

Reviewer's Responses to Questions

**Comments to the Author**

Reviewer #1: All comments have been addressed

Reviewer #2: All comments have been addressed

2. Is the manuscript technically sound, and do the data support the conclusions?

Reviewer #1: Yes

Reviewer #2: Yes

3. Has the statistical analysis been performed appropriately and rigorously?

Reviewer #1: Yes

Reviewer #2: Yes

4. Have the authors made all data underlying the findings in their manuscript fully available?

Reviewer #1: Yes

Reviewer #2: Yes

5. Is the manuscript presented in an intelligible fashion and written in standard English?

Reviewer #1: Yes

Reviewer #2: Yes

Reviewer #1: The authors have addressed all the comments thoroughly and have provided clear responses to every point raised

Reviewer #2: The histology images and immunohistochemistry (IHC) data, as in-situ analyses, could be incorporated into the main figure. Several of the cell line proliferation images appear redundant and lack sufficient resolution. I believe the authors have not sufficiently emphasized the significance of their findings. The manuscript and the evidence derived from negative research findings should be presented with the same rigor as positive results. This principle has been highlighted by other authors and serves to encourage researchers to report scientifically sound negative findings. The final image panels, as currently constructed, lack visual impact. I recommend reorganizing these panels by incorporating the new data presented in the supplementary materials. I leave the final decision to editorial board on this.

.

Reviewer #1: No

Reviewer #2: No

---

## [Author Response · Author response to Decision Letter 2]

2 Mar 2026

Point to point Reply

Reviewer 1

Comment 1:

Reviewer #1: The authors have addressed all the comments thoroughly and have provided clear responses to every point raised

Answer to Comment 1:

We thank the reviewer for his assessment of the manuscript.

Reviewer 2:

Comment 1:

Reviewer #2: The histology images and immunohistochemistry (IHC) data, as in-situ analyses, could be incorporated into the main figure. Several of the cell line proliferation images appear redundant and lack sufficient resolution. I believe the authors have not sufficiently emphasized the significance of their findings. The manuscript and the evidence derived from negative research findings should be presented with the same rigor as positive results. This principle has been highlighted by other authors and serves to encourage researchers to report scientifically sound negative findings. The final image panels, as currently constructed, lack visual impact. I recommend reorganizing these panels by incorporating the new data presented in the supplementary materials. I leave the final decision to editorial board on this.

Answer to Comment 1:

According to the reviewer’s suggestions, we have incorporated the histological analysis of the tumors into the revised manuscript as main Figure 4.

In addition, we adjusted the contrast of the cell line images to improve clarity. The resolution of these images in the TIFF- files is high to ensure optimal image quality.

To improve the overall structure and the readability of the manuscript, Figure.1 now presents only the proliferation and cell death analysis from the Panc02 cells, which were subsequently used for the western blot analysis (Fig. 2) and the orthotopic murine cancer model (Fig. 3 and 4).

The proliferation data from the other murine and humane cell lines are now provided in Supplementary Figure 1. (Please see the attached PDF-file)

---

## [Decision Letter · Decision Letter 2]

8 Apr 2026

Targeting pancreatic cancer with combined inhibition of EGFR and RAF

PONE-D-25-49774R2

Dear Dr. Kumstel,

We’re pleased to inform you that your manuscript has been judged scientifically suitable for publication and will be formally accepted for publication once it meets all outstanding technical requirements.

Kind regards,

Yoshihiro Sowa

Academic Editor

PLOS One
---

## [Editor Report · Acceptance letter]

PONE-D-25-49774R2

PLOS One

Dear Dr. Kumstel,

I'm pleased to inform you that your manuscript has been deemed suitable for publication in PLOS One. Congratulations! Your manuscript is now being handed over to our production team.

Kind regards,

on behalf of

Dr. Yoshihiro Sowa

Academic Editor

PLOS One